# A Survey of the Dung-Dwelling Arthropod Community in the Pastures of the Northern Plains

**DOI:** 10.3390/insects15010038

**Published:** 2024-01-06

**Authors:** Ryan B. Schmid, Kelton D. Welch, Jonathan G. Lundgren

**Affiliations:** Ecdysis Foundation, Estelline, SD 57234, USA; kelton.welch@ecdysis.bio (K.D.W.); jonathan.lundgren@ecdysis.bio (J.G.L.)

**Keywords:** dung-feeding beetles, biodiversity, rangeland, functional group

## Abstract

**Simple Summary:**

The European settlement of North America has led to dramatic changes in the environment of the Northern Plains. As this ecosystem’s plant and animal communities continue to evolve during this period of human-driven modification, it is important to continually measure the impact on important ecological groups in this system, such as dung-dwelling arthropods. Therefore, we report the dung-dwelling arthropod community collected from 40 pastures extending from northeast South Dakota to central North Dakota during the 2019 and 2020 grazing seasons. A total of 51,283 specimens were collected. Beetles, flies, and parasitoid wasps comprised the majority (94.5%) of the community. Pest abundance was low on average, with about one adult pest found for every two dung pats sampled, and most of the pats (80%) did not contain any adult pests. While there were many beneficial arthropods collected for this survey, the abundance of these beneficial species was inconsistent from one pat to the next and throughout the grazing seasons. Future work is needed to understand how to increase the consistency of these beneficial organisms in the grassland ecosystem of the Northern Plains.

**Abstract:**

Grassland ecosystems of the Northern Plains have changed substantially since European settlement began in the latter half of the 19th century. This has led to significant changes to the dung-dwelling arthropod community in the region. As humans continue to modify large portions of the landscape, inventories of ecologically significant communities are important to collect in order to monitor the long-term effects of anthropogenic biomes. We conducted a survey of the arthropod community dwelling in cattle dung from 40 pastures extending from northeast South Dakota to central North Dakota during the 2019 and 2020 grazing seasons. In sum, 51,283 specimens were collected from 596 dung pats, comprising a community of 22 orders. Coleoptera, Diptera, and Hymenoptera contributed to the majority (94.5%) of the community abundance. The mean pest abundance was low per pat (0.43 adult pests/pat), with 80% of the pats not containing any adult pest. Ecologically beneficial dung-feeding beetles, predators, and parasitoids were abundant in the region, but it was an inconsistent community, which may hinder ecosystem services. This highlights the need for future work to understand the mechanisms to increase the consistency of dung pat colonization for improved consistency of ecosystem services in the region.

## 1. Introduction

The Northern Plains of the United States is home to vast numbers of large ruminants, with historical herds largely comprising bison, deer, elk, and pronghorn that today have been mostly replaced by another large ruminant, domestic cattle [1]. The introduction of European cattle to North America was accompanied by the introduction of numerous dung beetle species [2]. These introduced species added to what was already considered a diverse dung beetle community [3,4]. The current dung community has evolved alongside the changes in the plant communities of the northern Great Plains, associated with row crop and livestock production, and changes in livestock management intensities; both factors are known to affect dung arthropod community composition [5,6,7,8,9]. As this region continues to reckon with increased agriculture intensification, accompanied by shifting weather patterns due to climate change, surveys of the dung arthropod community support studies examining the changes in this community over time. Dung arthropod surveys also inform present-day research exploring the capacity of the Northern Plains’ dung arthropod community to perform important ecosystem services for ranchers, e.g., recycling dung into the soil.

We sampled the dung arthropod community in the region, recognizing that this community may still be evolving after the introduction of new dung beetles in the region, which has been accompanied by an increasing intensification of agriculture. Therefore, the primary objective of this study was to provide a record of the dung arthropod fauna from a region of the Northern Plains extending from northeast South Dakota to central North Dakota. Secondarily, we examined the distribution of functional groups within the dung fauna.

## 2. Materials and Methods

### 2.1. Study Sites 

Sampling of arthropod communities in cattle dung occurred within roughly an 80 km wide band that extended from northeast South Dakota/west central Minnesota to central North Dakota (Figure 1). Selected sites (n = 40) were pastures used for cattle grazing and were managed by 36 ranchers. Consequently, a variety of pasture and livestock management strategies were applied to the pastures that were sampled. Management systems had been practiced on pastures for at least 4 years prior to sampling. Sampling of dung-dwelling arthropods took place three times (early June, mid July, late August/early September) during both the 2019 and 2020 grazing seasons, with half the sites sampled in 2019 and the other half sampled in 2020.

### 2.2. Sampling Procedure 

Dung-dwelling arthropod communities were sampled via inserting a core sampling cutter (10 cm diameter, 10 cm deep) through the center of a dung pat. The age of dung pats selected for sampling was between 2 and 4 d old, as this age of pat has the peak arthropod abundance and diversity in the region [10]. Pat age was determined through experiential knowledge of sampling dung pats at this age in previous experiments, and this experience was calibrated each day of sampling by observing dung pats in paddocks that cattle had been released into within the previous 2–4 days. Dung cores were kept on ice upon extraction from the field until they could be returned to the laboratory. Cores were then placed in a Berlese funnel system for 10 d to ensure cores had completely dried and all arthropods had evacuated. Arthropods extracted from cores were stored in 70% isopropyl alcohol until they could be identified and cataloged.

### 2.3. Community Composition 

Specimens were identified to the lowest taxonomic level possible using peer-reviewed taxonomic keys and consultation of taxonomic experts, as needed. Owing to a lack of taxonomic keys, available expertise, and time constraints, the following arthropod groups were only identified to the respective levels: mites (Arachnida: Acari) identified to the order level, Protura identified to the class level, thrips (Insecta: Thysanoptera) identified to the order level, springtails (Hexapoda: Collembola) identified to the family level, and immature specimens to order or family level. These insect groups were excluded from all data metrics (abundance, species richness, diversity, and functional groups) because their inclusion would confuse the interpretation of the community owing to our inability to identify the specimens to distinct morphospecies. All other specimens were identified to genus and species level or assigned a unique morphospecies number. Functional groups were assigned to morphospecies of ecologically important insect orders dwelling in cattle dung (Araneae, Coleoptera, Diptera, and Hymenoptera) citing literature and current hypotheses of the ecology of these organisms. Functional groups assigned to these orders included coprophagous, mycophagous, parasitoid, and predator. While coprophagous beetles typically only refer to beetles belonging to the Scarabaeoidea, we included coprophagous beetles from the family Hydrophilidae in this functional group, as many adult Hydrophilidae beetles feed on dung in the region. Also, the coprophagous beetle functional group was broken down further into dweller, roller, and tunneler, as this is commonly accepted terminology in the peer-reviewed literature and vernacular used by the ranching community at the time of publication of this paper. However, the authors recognize that Tonelli [11] recently proposed a revision of the dweller group based on nesting behavior, calling for the distinction between non-nesting and endocoprid species. Therefore, we recognized this distinction when discussing the data. Voucher specimens were deposited in the Mark F. Longfellow Ecological Reference Collection, housed at Blue Dasher Farm (Estelline, South Dakota, USA).

### 2.4. Data Analysis 

Arthropod morphospecies were enumerated to calculate arthropod abundance, species richness, Hill species diversity metrics (exponential Shannon–Wiener H’, inverse Simpson, and inverse Berger–Parker), and species evenness (Shannon equitability). Morphospecies assigned to functional groups within the orders Araneae, Coleoptera, Diptera, and Hymenoptera were tallied to present abundance and species richness of each functional group.

## 3. Results

### 3.1. Arthropod Community

A total of 51,283 arthropods specimens were collected from 596 dung pats from 40 pastures throughout the 2019 and 2020 grazing seasons in North Dakota, South Dakota, and Minnesota. This dung arthropod community was represented by 787 morphospecies from seven classes (Arachnida, Chilopoda, Diplopoda, Diplura, Insecta, Malacostraca, and Symphyla) and 21 orders (Araneae, Coleoptera, Dermaptera, Dicellurata, Diptera, Geophilomorpha, Hemiptera, Hymenoptera, Isopoda, Julida, Lepidoptera, Lithobiomorpha, Neuroptera, Opiliones, Polydesmida, Pseudoscorpiones, Psocodea, Psocoptera, Rhabdura, Symphyla, and Trichoptera). Mites, Collembola, Protura, Thysanoptera, and immature specimens were also found in the samples, but these arthropods were not included in the results owing to lack of taxonomic keys and time to identify these organisms to morphospecies. A complete list of arthropod specimens comprising these results and their abundance is provided in the Appendix A accompanying this article.

Three arthropod orders, Coleoptera, Diptera, and Hymenoptera, comprised the majority (94.5%; n = 48,463 specimens) of the arthropod community abundance (Figure 2A). The remaining 5.5% of the arthropod community was distributed among the remaining 18 orders. The mean arthropod abundance per pat varied within and between sampling seasons (Figure 3A). In the 2019 grazing season, arthropod abundance peaked early in the season (103.3 ± 8.9), with the lowest abundance occurring during the middle of the season (82.0 ± 6.0) before rebounding at the end of the season (94.3 ± 7.7). By comparison, the early portion of the 2020 grazing season had the lowest overall abundance (49.3 ± 6.3) of any of the sampling points before spiking to the highest abundance in the middle of the grazing season (105.9 ± 5.4).

The species richness of the community was consistent with the abundance results. Diptera, Coleoptera, and Hymenoptera were the three most species-rich orders, comprising 77.9% of the species, with the remaining 22.1% of the species distributed across the remaining 18 orders (Figure 2B). Mean species richness per pat fluctuated during the grazing seasons, with a general decline from early to late in the 2019 grazing season (early: 25.5 ± 1.1, mid: 18.1 ± 0.7, late: 16.3 ± 0.7), while peak species richness occurred in the middle of the 2020 grazing season (early: 17.6 ± 0.9, mid: 27.4 ± 0.7, late: 13.2 ± 0.7) (Figure 3B). Hill species diversity metrics (Table 1) indicated that the fluctuations in arthropod abundance and species richness observed in the middle portion of the 2020 season were driven by an increase in rare species. The exponential Shannon–Wiener index (Figure 3C), which is sensitive to rare species, showed an increase in diversity during the middle portion of the 2020 season, while the diversity indices that emphasize dominant species, Simpson and Berger–Parker, showed a steady decline in diversity through the 2020 season, which was the same as in the 2019 season. Species evenness reflected the Simpson and Berger–Parker diversity indices, with both the 2019 and 2020 grazing seasons showing a steady decline in evenness from the early to the late sampling periods (Figure 3D).

### 3.2. Functional Groups

Out of the 210 Coleoptera morphospecies identified for this study, 169 were assigned to a functional group. Predators constituted the largest proportion, 41% of beetle community abundance (n = 10,904 specimens), comprising 111 species. These were followed by coprophagous and mycophagous beetles, 36% and 23%, respectively (Coprophagous: 9,572 specimens, 27 species; Mycophagous: 6,214 specimens, 42 species) (Figure 4A,B). Of the coprophagous beetles, nearly all were dwellers (9,151 specimens, 22 species) with a few tunnelers (421 specimens, 5 species), and no rollers were collected. All dweller species collected were non-nesting, with no endocprids. It should be noted that *Teuchestes fossor* was labeled as a dweller in the dataset, despite conflicting observations stating this species is a dweller and a tunneler [12,13]. This species was labeled a dweller in this dataset for two reasons 1) Gittings and Giller [13] made direct observations of *T. fossor* oviposition and larval development in dung pats, and 2) the source observing *T. fossor* to be a tunneler states that the third instar larval stage, which is the longest developing stage, develops in the dung pat [12]. Should readers consider *T. fossor* to be a tunneler, their abundance was 226 specimens, and the following data can be adjusted accordingly. The mean number of dung-feeding beetles per dung pat was 16.1, but the number of dung-feeding beetles per dung pat varied considerably among pats throughout each grazing season (Figure 5A). The dung-feeding beetle abundance in pats ranged from 0 to 130, with a median of 10 beetles.

In total, 132 of the 281 Diptera morphospecies were assigned to a functional group. Almost the entire Diptera community assigned a functional group were coprophagous, 99.7% (19,089 specimens, 114 species), relative to predators (52 specimens, 19 species) and parasitoids (5 specimens, 4 species) (Figure 4A,B). Of the 22,014 Diptera collected in this survey, only 254 were pests, constituting 1.2% of the Diptera population. The pest community comprised four species: *Stomoxys calcitrans* (L.) (Diptera: Muscidae), *Haematobia irritans* (L.) (Diptera: Muscidae), *Musca domestica* L. (Diptera: Muscidae), and *Musca autumnalis* De Geer (Diptera: Muscidae). The mean abundance of Diptera pests per dung pat sampled was 0.43, with 477 out of 596 pats not having any pests detected, and the highest abundance of pests in a single pat being 27 (Figure 5D). This resulted in a median of 0 pests per pat.

In addition to Coleopteran and Dipteran functional groups, other orders of arthropods contributed significant abundance to the parasitoid and predator community. Araneae contained 131 predator specimens (14 species) and Hymenoptera contributed 546 parasitoid specimens (109 species). With the addition of these predators and parasitoids, the mean abundance of arthropod predators and parasitoids per dung pat was 18.6 and 0.9, respectively, but abundances varied substantially between dung pats (Figure 5B,C). Predators varied from 0 to 111 specimens per pat, with a median of 11, while parasitoids varied from 0 to 14 specimens per pat, with a median of 0.

Collectively, 78% of the community was assigned to a functional group, of which there were 28,661 coprophagous, 6,214 mycophagous, 11,087 predators, 551 parasitoids, and 254 pests. On a per-pat basis, this was equivalent to 48.1 coprophagous, 10.4 mycophagous, 18.6 predators, 0.9 parasitoids, and 0.43 pests per pat.

## 4. Discussion

This study described the dung arthropod community extending from northeast South Dakota/west central Minnesota to central North Dakota. The sampled community was primarily comprised of Coleoptera, Diptera, and Hymenoptera, which represented 94.5% of the abundance (n = 48,463 specimens) and 77.9% of the species (n = 613 species). The remainder of the arthropod community was distributed across 174 species, spanning 18 orders. These results align with descriptions of dung arthropod communities of other north temperate regions, mostly comprising a mixture of coprophagous beetles and dung flies, accompanied by predatory and parasitic mites, beetles, flies, and wasps [14]. Coprophages were the most abundant functional group from our survey, with Diptera contributing 19,089 individuals and Coleoptera contributing 9,572 individuals, totaling 28,661 coprophages sampled from 596 dung pats (Figure 4A). This produced a mean of 48.1 dung-feeding arthropods per dung pat (32.0 dung-feeding flies and 16.1 dung-feeding beetles per pat). Predators and parasitoids (n = 11,093 predators, 551 parasitoids) also contributed a large portion of the sampled community, with a mean of 18.6 predators and 0.9 parasitoids per pat. Despite the relatively robust beneficial arthropod community that these averages portray, each of the aforementioned functional groups displayed a significant amount of variability among dung pats (Figure 5). The inconsistent colonization of pats by these important functional groups may result in the unreliable provisioning of ecosystem services, such as pest control and dung degradation.

In terms of dung degradation in north temperate ecosystems, it is estimated that only 1–5% of the dung material leaves the pat via insects, but insects contribute significantly to the initial breakup of the pat, allowing for further colonization by other organisms [15]. Most of this degradation is provided by dung-feeding beetles, as coprophagous fly larvae consume primarily a liquid diet and do not eat enough organic matter to have a significant effect on dung degradation [16]. Beetle size, nesting habits, and reproduction cycle are just a few of the morphological and life cycle attributes of dung-feeding beetles that affect dung removal rates [17,18]. In general, tunnelers and rollers have larger body sizes and nesting habits that remove and bury larger quantities of dung into the soil, and, thus, are more efficient at reincorporating dung into the soil than smaller dweller species (like Aphodiinae) that feed within the dung pat [19,20]. Our sampled arthropod community was largely devoid of the more efficient dung-degrading roller and tunneler specimens, with dwellers representing 98.5% of the community, represented entirely by non-nesting species. However, it should be noted our sampling method was biased against the collection of rollers and tunnelers because these two groups of dung beetles only spend a brief period of time in the dung pat. Still, high abundances of smaller dweller dung-feeding beetles have been observed in the region earlier in the 20th century, resulting in quick pat degradation [5,21]. This demonstrates that the overall biomass of dung-feeding beetles present in a pat can be just as important for rapid dung degradation as the presence of a few large tunneler or roller species [22]. However, we did not find a high abundance of dwellers comparable to previous observations that resulted in quick dung pat degradation. This result, together with the lack of rollers and tunnelers collected with our sampling method, indicates that the dung-feeding beetle community in the region may not be capable of rapid dung degradation, relative to those in other regions of the continent that have greater abundances of tunnelers and rollers [23,24]. Thus, further research is warranted to (1) assess the full suite of functional dung beetle groups using an appropriate method to document the abundance of rollers and tunnelers in the region and (2) determine if the dung community in this region of the United States is capable of performing the important ecological function of dung recycling. 

Dung degradation is not the only ecosystem service provided by dung-dwelling arthropods. Dung fauna also help control the populations of dung-dwelling fly pests and parasites of livestock. This service is achieved through a combination of functional groups: directly by predators and parasitoids and indirectly by dung-feeding arthropods through the consumption of pest nesting habitat. The potential lack of rapid pat degradation by the surveyed dung beetle community was already discussed, but, in addition to dung beetles, we did find mean numbers of predators (18.6 per pat) and parasitoids (0.9 per pat), while inconsistent, to be a large proportion of the dung arthropod community. The question then becomes, are these levels of predators and parasitoids high enough to control pests? The survey showed pest populations to be consistently low in the sampled dung pats, with a mean abundance of 0.43 pests per pat and 477 out of 596 pats containing no adult fly pest. This shows that adult fly pests were found in low abundance in our sampled pats. However, this does not necessarily mean that predators, parasitoids, and dung-feeding arthropods were the only factors contributing to fly pest control. Insecticides are a common fly control tactic used by ranchers in the region, with 20 out of 36 participating ranchers using insecticides immediately prior (≤1 month) or during the grazing season. Because insecticide regiments were implemented according to each ranch’s specific pest management program, there were a variety of insecticide delivery mechanisms, modes of action, and rates applied to the different herds across the ranches. This resulted in no replication amongst pasture sites to disentangle the effect that beneficial arthropods had on pest control apart from insecticides. However, of the pats that contained fly pests, 59% of pest abundance was from herds that had insecticide applications immediately prior to and/or during the grazing season. Furthermore, over half the pest abundance came from just 18 pats, 66% of which were from insecticide -treated herds, and three of these insecticidal pats harbored nearly a quarter of the pest abundance. This indicates that when pats did contain pests in high abundance, they tended to be from herds that received insecticides immediately prior to and/or during the grazing season. This observation calls for further research to assess the economic value of insecticides versus a healthy, well-balanced arthropod community for fly pest control. 

This survey occurred at a pivotal time in terms of landscape transformation in the region. Most of the grasslands of the eastern Dakotas have been converted to row crop production following European settlement, with this trend intensifying during the 21st century on the few remaining grasslands in the region [25,26]. As the proportion of the landscape dedicated to the production of large herbivorous herds declines and becomes increasingly fragmented, surveys like this serve as an important waypoint for future research in this rapidly changing region. This survey shows a dung-dwelling arthropod community that closely reflects those found in other northern temperate regions [23]. Coprophagous beetles and flies, along with predatory and parasitic beetles, flies, and wasps, were the most abundant functional groups present. Although, even this small cadre of functional groups shows that changes are happening in the community. For example, the majority of the coprophagous beetle species collected were non-native (n = 17 non-native, n = 10 native), which may lead to shifts in nutrient cycling efficiency relative to the native dung beetle community prior to European settlement. This is just one example that needs continued monitoring to understand how grassland loss and fragmentation are affecting the ecosystem services in the region 

## Figures and Tables

**Figure 1 insects-15-00038-f001:**
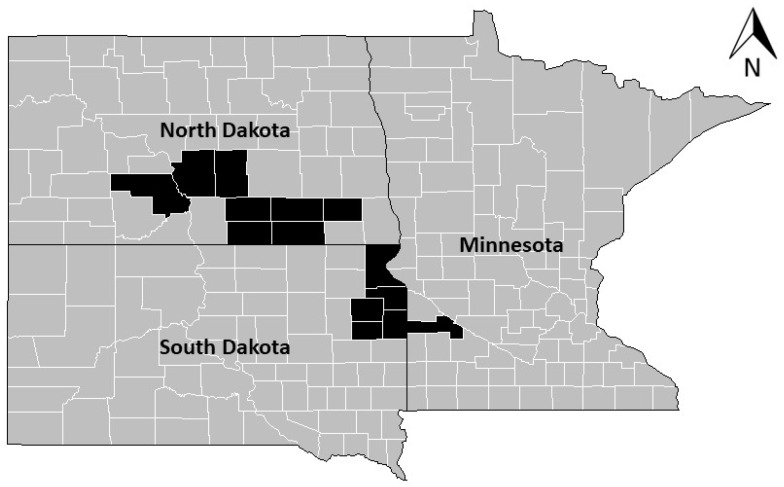
Pastures (n = 40) sampled for this study were located in Yellow Medicine County, MN; Burleigh County, ND; Dickey County, ND; Kidder County, ND; La Moure County, ND; Logan County, ND; McIntosh County, ND; Morton County, ND; Ransom County, ND; Codington County, SD; Deuel County, SD; Grant County, SD; Hamlin County, SD; and Roberts County, SD. Counties where sampling occurred are highlighted in black on state county maps.

**Figure 2 insects-15-00038-f002:**
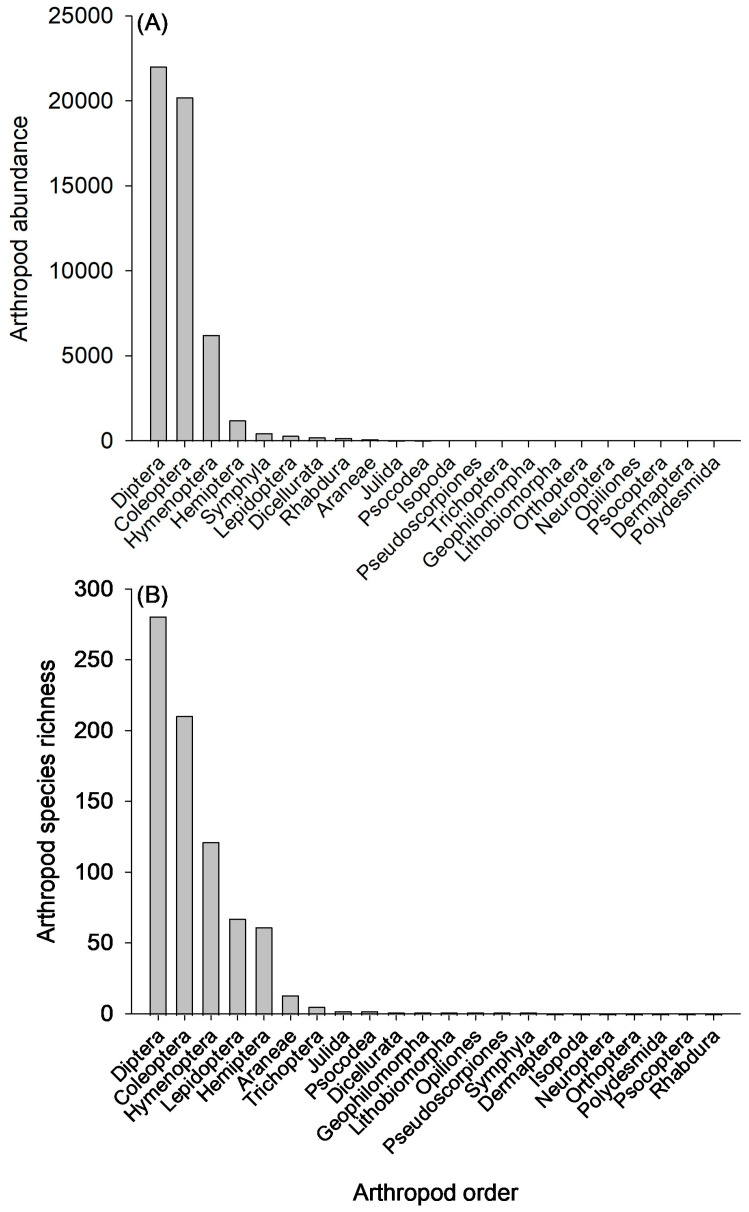
Overall (**A**) abundance and (**B**) species richness of arthropod orders collected during the survey.

**Figure 3 insects-15-00038-f003:**
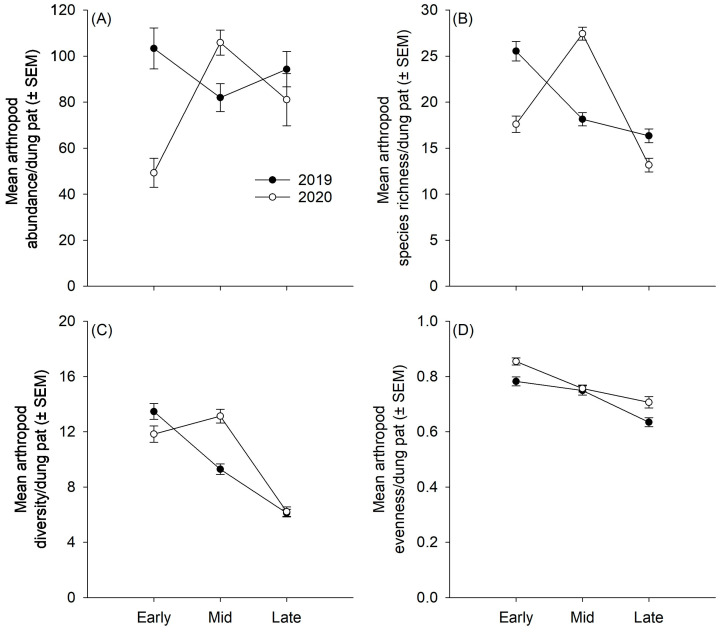
Mean (± SEM) arthropod (**A**) abundance, (**B**) species richness, (**C**) species diversity (exponential Shannon–Wiener H’), and (**D**) species evenness (Shannon equitability) per dung pat spanning the grazing season (early, mid, and late) during the 2019 and 2020 grazing seasons. Specifically, early sampling took place 12–28 June in 2019 and 2–11 June in 2020, mid-sampling took place 25 July–7 August in 2019 and 14–23 July in 2020, and late sampling took place 27 August–12 September in 2019 and 25 August–3 September in 2020.

**Figure 4 insects-15-00038-f004:**
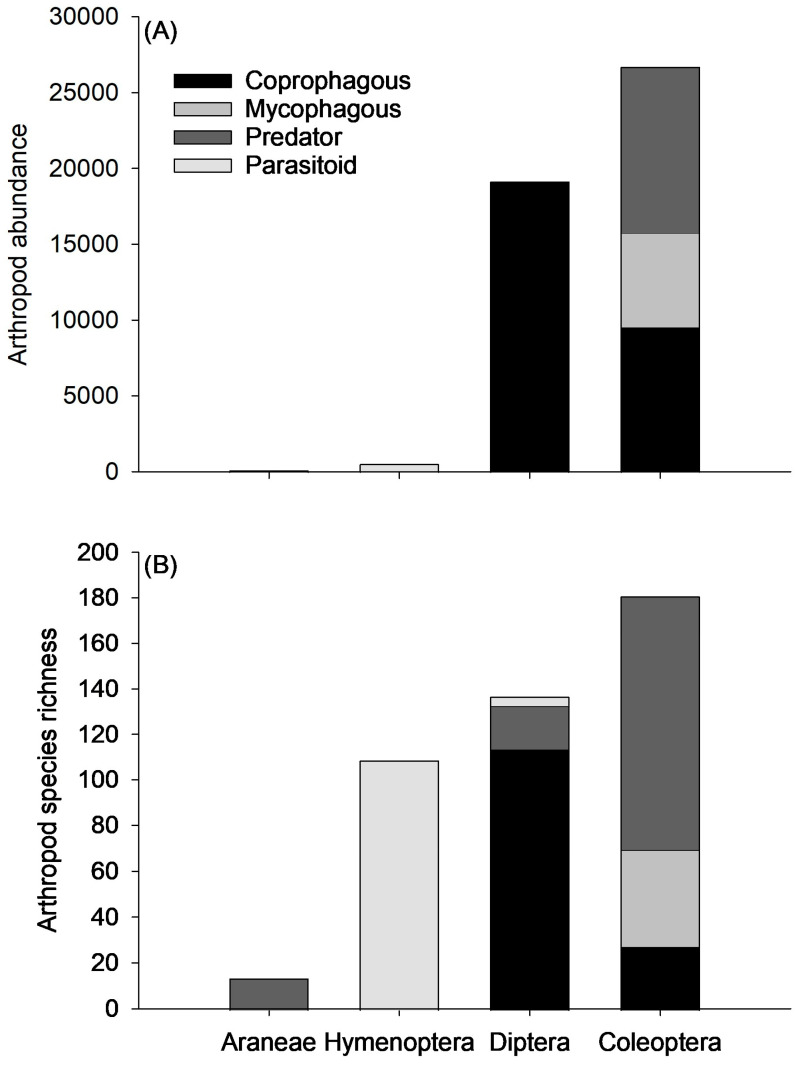
(**A**) Abundance and (**B**) species richness of specimens categorized into functional groups within the arthropod orders Araneae, Hymenoptera, Diptera, and Coleoptera. Quantity of morphospecies categorized in each order were as follows: 14 of 14 Araneae morphospecies, 109 of 122 Hymenoptera morphospecies, 169 of 210 Coleoptera morphospecies, and 132 of 281 Diptera morphospecies. Only the functional groups coprophagous, mycophagous, predator, and parasitoid were applied to these arthropod orders.

**Figure 5 insects-15-00038-f005:**
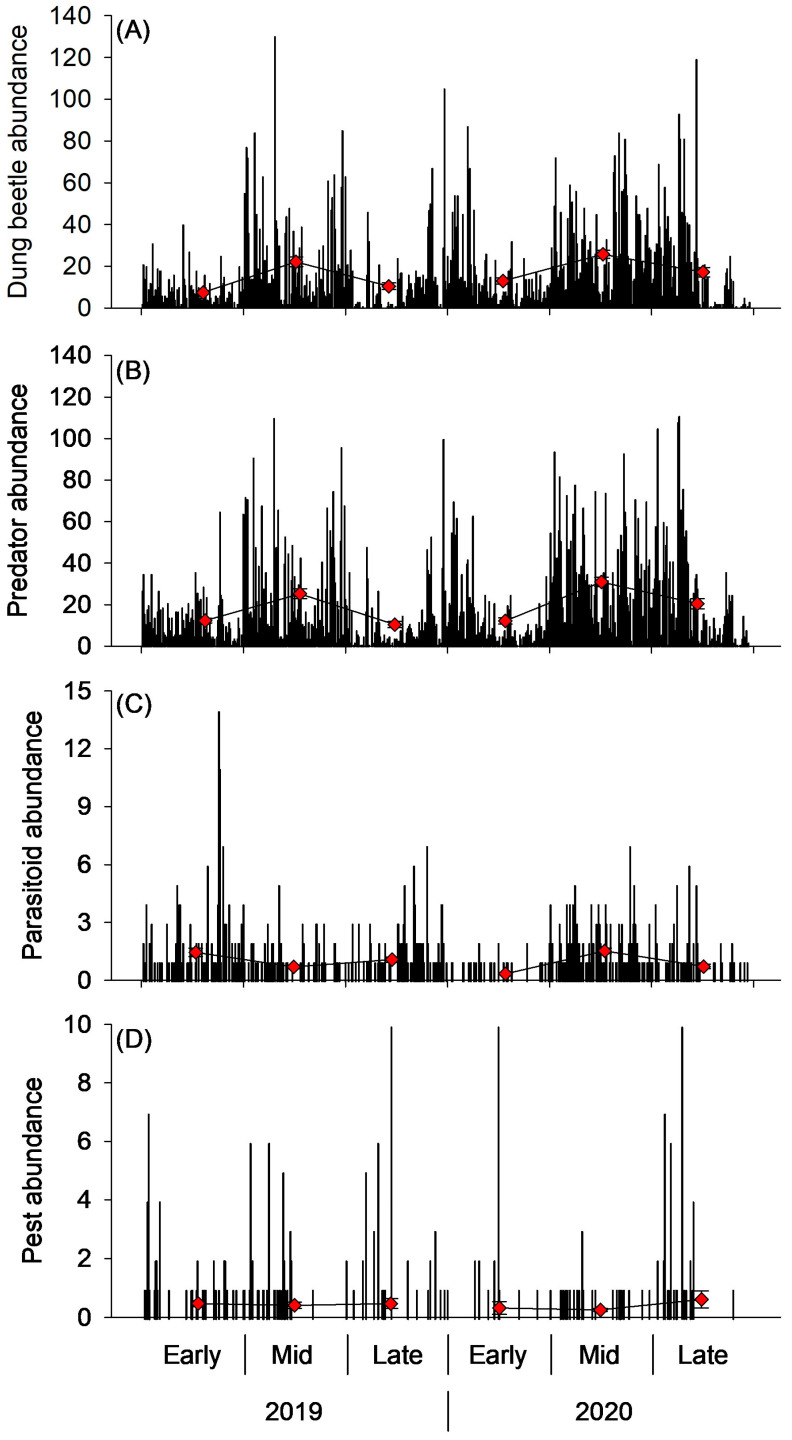
Abundance of (**A**) dung-feeding beetles, (**B**) predators, (**C**) parasitoids, and (**D**) pests per dung pat. Each bar represents one sampled dung pat. Pats were sampled during the early, mid, and late periods of the 2019 and 2020 grazing seasons. Mean abundance (±SEM) is represented with red diamonds. Specific sampling dates were 12–28 June 2019 and 2–11 June 2020 (early), 25 July–7 August 2019 and 14–23 July 2020 (mid), and 27 August–12 September 2019 and 25 August–3 September 2020 (late).

**Table 1 insects-15-00038-t001:** Hill species diversity metrics of dung arthropod community sampled from northeastern South Dakota to central North Dakota. Columns dictate mean (±SEM) arthropod species richness (Hill number N0), exponential Shannon–Wiener diversity (Hill number N1), inverse Simpson diversity (Hill number N2), and inverse Berger–Parker diversity (Hill number N3).

Year	Season	Species Richness	Exponential Shannon-Wiener	Inverse Simpson	Inverse Berger-Parker
2019	Early	25.5 ± 1.1	13.4 ± 0.6	10.7 ± 0.6	4.4 ± 0.2
	Mid	18.1 ± 0.7	9.3 ± 0.4	7.3 ± 0.4	3.2 ± 0.1
	Late	16.3 ± 0.7	6.1 ± 0.3	3.8 ± 0.2	2.1 ± 0.1
2020	Early	17.6 ± 0.9	11.8 ± 0.6	15.5 ± 1.8	4.4 ± 0.2
	Mid	27.4 ± 0.7	13.1 ± 0.5	9.2 ± 0.6	3.9 ± 1.2
	Late	13.2 ± 0.7	6.2 ± 0.3	5.3 ± 0.5	2.7 ± 0.1

## Data Availability

The data presented in this study are available in the Appendix A.

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
