# Peer review of "A Survey of the Dung-Dwelling Arthropod Community in the Pastures of the Northern Plains"

_insects, 2024, doi:10.3390/insects15010038_

Round 1
Reviewer 1 Report
Comments and Suggestions for Authors
The authors have conducted an important survey of dung dwelling arthropods on cattle ranches in the Northern Great Plains of North America. The key result, that pest flies were only found in 20% of dung pats shows that the resource of cattle dung supports a large number of non-pest invertebrate fauna. Overall, I found the study to be an important contribution. I do have some small concerns:
1) While the methods are sufficient for collecting invertebrates that dwell in dung, there are also invertebrates that use but do not dwell in dung. These include the two functional guilds of dung beetles of rollers and tunnellers. The methods of this study are biased against those guilds. The Discussion in particular needs to incorporate this fact and not state that these guilds are missing when they are likely just not being sampled.
2) The Discussion is primarily focused on dung beetles. It would be nice to have more interpretation of the results of other taxa.
3) The authors end by stating that the low pest fly abundance is caused by rancher management but there is no data to back up this statement. Either this should be removed or management/treatment data should be additionally analyzed.
Minor comments
Line 73: How did the authors know that a dung pat was 2-4 days old? Were pats observed when they were deposited?
Line 80 and below: I would prefer “Order level” to “ordinal level”. Ordinal commonly refers to order in another sense- that of a ranking- and is a bit confusing to use in the context of taxonomical hierarchy.
Lines 116 -121: Why did 2019 and 2020 have different signals of abundance across the growing season? Were there major weather differences between years?
Lines 147-148: Small caveat of this study that the methods were not ideal for catching roller dung beetles and likely a bias against tunneller as well. It would be good to mention this in the Discussion (e.g. around line 202 when discussing geography of dung beetle fauna and has larger implications for lines 235-237: I think the interpretation that there are few rollers incorrect- they just spend much less time in pats)
Lines 191-192: Can delete the part about the three orders which is repetitive of above
Discussion/Results: What is the percent of dung beetles that were native v non-native cattle associates?
Lines 223- 234: This section is very results heavy and does not have much “discussion”. Would be nice to expand this section a bit re importance of these other guilds as had been done for dung beetles. For example, could any of the parasitoids be natural enemies of the pest flies?
Line 267: The data does not show that ranchers are the cause of low pest flies, just that there are low pest flies. Any data on how ranching management drove variation in insect communities would be an excellent addition but is probably outside of the scope of the what that authors can add at this stage. I recommend to weaken the strength of the statement.
Comments on the Quality of English Language
The quality of English is good.
Author Response
It is clear Reviewer 1 took the time to thoroughly review our manuscript and provide constructive feedback. We thank the reviewer for their time and helpful comments. We have done our best to incorporate reviewer suggestions into the manuscript or noted otherwise. Please find below our responses to reviewer comments.
Sincerely,
Ryan Schmid
Kelton Welch
Jon Lundgren
Reviewer 1 Comments
The authors have conducted an important survey of dung dwelling arthropods on cattle ranches in the Northern Great Plains of North America. The key result, that pest flies were only found in 20% of dung pats shows that the resource of cattle dung supports a large number of non-pest invertebrate fauna. Overall, I found the study to be an important contribution. I do have some small concerns:
1) While the methods are sufficient for collecting invertebrates that dwell in dung, there are also invertebrates that use but do not dwell in dung. These include the two functional guilds of dung beetles of rollers and tunnellers. The methods of this study are biased against those guilds. The Discussion in particular needs to incorporate this fact and not state that these guilds are missing when they are likely just not being sampled.
We agree that our sampling method is bias against rollers and tunnelers because these functional guilds only spend a brief period of time in the dung pat. We have included this point in the Discussion. (Lines 267-270, 343-350)
2) The Discussion is primarily focused on dung beetles. It would be nice to have more interpretation of the results of other taxa.
This is a valid point. We have substantially revised the Discussion in accordance to several comments from both reviewers. One of the changes made was discussion of the other functional groups and the potential effects of the changing landscape in the region on the dung arthropod community. (Lines 351-395)
3) The authors end by stating that the low pest fly abundance is caused by rancher management but there is no data to back up this statement. Either this should be removed or management/treatment data should be additionally analyzed.
We have revised this section of the Discussion owing to the reviewer’s comment. Our revisions present the potential for the arthropod community and/or insecticide treatments as two likely candidates for the low pest abundance, but we are not able to determine the cause with the dataset and this requires further research to determine the cause. We feel this is an important observation to make in the Discussion based on the survey results. (Lines 351-379)
Minor comments
Line 73: How did the authors know that a dung pat was 2-4 days old? Were pats observed when they were deposited?
We did not observe pats as they were deposited. However, we do have previous experience sampling pats of this age, and we also had cooperating ranchers that were rotating cattle through paddock systems frequently enough, that we could observe dung pats that were deposited 2-4 days prior during each of our sampling days. This let us calibrate our search image for pats in this age range. We have added a line to the manuscript to clarify this point to the reader. (Line 78)
Line 80 and below: I would prefer “Order level” to “ordinal level”. Ordinal commonly refers to order in another sense- that of a ranking- and is a bit confusing to use in the context of taxonomical hierarchy.
Agreed. We have made the requested edits throughout the manuscript.
Lines 116 -121: Why did 2019 and 2020 have different signals of abundance across the growing season? Were there major weather differences between years?
We are unsure what caused the significant differences in abundance between the two seasons. It could have been related to several factors including weather or the fact that different pastures were sampled between the two seasons, which opens several other factors that may have affected the abundances, like surrounding landscape, livestock/pasture management, latitude or longitude of the pastures, etc. Because it is nearly impossible to narrow down the cause of the differences between the two seasons, we have chosen to simply present the data as-is to the reader but not discuss it further, as it would merely be speculation without a clear testable outcome for future research. This would seem like an unnecessary distraction to insert in the Discussion section.
Lines 147-148: Small caveat of this study that the methods were not ideal for catching roller dung beetles and likely a bias against tunneller as well. It would be good to mention this in the Discussion (e.g. around line 202 when discussing geography of dung beetle fauna and has larger implications for lines 235-237: I think the interpretation that there are few rollers incorrect- they just spend much less time in pats)
Agreed. We have included this in the Discussion as part of a larger expansion of the subject, as suggested by Reviewer 2. (Lines 267-270, 343-350)
Lines 191-192: Can delete the part about the three orders which is repetitive of above
This line has been deleted. (Line 240)
Discussion/Results: What is the percent of dung beetles that were native v non-native cattle associates?
17 of the 27 dung beetle species were non-native. This information has been added in the Discussion section. (Line 391)
Lines 223- 234: This section is very results heavy and does not have much “discussion”. Would be nice to expand this section a bit re importance of these other guilds as had been done for dung beetles. For example, could any of the parasitoids be natural enemies of the pest flies?
Agreed. We have expanded our discussion on the other functional groups. (Lines 351-379)
Line 267: The data does not show that ranchers are the cause of low pest flies, just that there are low pest flies. Any data on how ranching management drove variation in insect communities would be an excellent addition but is probably outside of the scope of the what that authors can add at this stage. I recommend to weaken the strength of the statement.
Agreed. We have revised this paragraph to simply state the predator and parasitoid abundance, reported the low pest abundance, and then report pest abundance in relation to producers that used insecticides. We were careful to include a statement that this dataset cannot distinguish between the effects of insecticides and predators/parasitoids on the pest population. (Lines 351-379)
Reviewer 2 Report
Comments and Suggestions for Authors
The Authors sampled the dung arthropod community in the Northern plains region (South Dakota, Minesota and North Dakota) three times (June, July and August/September) during two grazing seasons (2019 and 2020).
They characterized the arthropod communities calculating abundance, species richness, species diversity (Shannon H), species evenness (Shannon equitability), functional group abundance and functional group species richness.
Their main results are:
- They collected 51,290 arthropod specimens, belonging to 789 morphospecies, from 596 dung pats from 40 pastures
- Diptera, Coleoptera and Hymenoptera represent the majority of arthropod community abundance (94.5%) and species richness (77.8%).
- Arthopod abundance and species richness seems do not show a clear pattern during the three period of sampling.
- Arthopod diversity and evenness showed a steady decline from early to late sampling period.
- Coprophagous and predators represent the majority of the functional groups abundance and species richness
Overall, I think that the data of the authors are interesting, especially due to the large spatial scale of the sampling. However, I think the data could be presented better. Indeed, my major concern are about the data presentation and several speculative points in the discussion.
Some possible suggestions to improve the manuscript are:
- Line 56: you use the words “trophic groups”, “functional group”, “guild”…please be consistent. They are not synonyms.
- Line 61: (Fig. 1)…. Where is Fig 1?
- Lines 90-91 and throughout the article and supplementary material: I think the authors have to change dwellers with non-nesters. Indeed, the category dweller include also endocoprid species which are not present in your area. I suggest to the authors to avoid the categories dwellers, tunnelers and rollers due to their ambiguity (especially regarding dweller category), and use the chategories telecoprid, paracoprid, endocoprid and non-nesters. I suggest to make this change throughout the manuscript. See Tonelli, M., 2021. Some considerations on the terminology applied to dung beetle functional groups. Ecological Entomology, 46: 772-776.
- Results: I would like to see some measure of the amount of variation of your data… in the text (e.g. when you show the mean number of specimens per dung pat) and in the figure (e.g. figure 3)
- Functional groups: you did not report the results regarding richness in the different functional groups.
- Functional groups: if you want to have an ecological functional approach, I think it is of little use to continue dividing specific richness and abundance into different taxonomic orders. I think it is better to merge all the data of the different taxa using only the functional groups as categories.
- Lines 154-155 (Figure 4): ….”citing literature and current hypotheses of the ecology of these organisms…” ????
- Figure 5: I think this is not the best way to show the variability of the data. Maybe a table with the variance of the data could be better.
- Lines 203-204: generally Hydrophilidae are not considered “dung beetles”. (Krell, F.-T. & Moon, A.R. 2019. Quick guides: Dung beetles. Current Biology 29 (12): R554‒R555)
- Lines 235-255: this paragraph seems too speculative. For example, Hirschberger studied only Aphodius ater, but you extrapolate this data to all the Aphodiinae. Moreover, your data of 15.8 dweller/pat consider also predator Hydrophilidae (in the suppl material predator Hydrophilidae are categorized also as coprophagous and dwellers)?
Lines 256-279: to make this part less speculative you could see if the pests are less in the dung pats where coprophagous fauna is present. Moreover, is true that insecticides can control pests, but it also has many negative effects on dung beetle biodiversity and their ability to degrade excrement (see for example Tonelli et al. 2020. Dung beetles: Functional identity, not functional diversity, accounts for ecological process disruption caused by the use of veterinary medical products. J. Insect Conserv., 24: 643–654. https://doi.org/10.1007/s10841-020-00240-4.
Verdù et al. 2018. Ivermectin residues disrupt dung beetle diversity, soil properties and ecosystem functioning: An interdisciplinary field study. Science of the Total Environment, 618: 219-228. https://doi.org/10.1016/j.scitotenv.2017.10.331)
- Supplementary Material: Colobopterus erraticus and Teuchestes fossor are paracoprid species (Rojewski, C., 1983. Observations on the nesting behaviour of Aphodius erraticus (L.). Polskie Pismo entomologiczne, 53: 271-279. ----- Zunino, M., Barbero, E., 1990. Food relocation and the reproductive biology of Aphodius fossor (L.) (Coleoptera Scarabaeidae Aphodiinae). Ethology Ecology & Evolution, 2: 334). Moreover, in the column EO the species name is missing.
Author Response
It is clear Reviewer 2 took the time to thoroughly review our manuscript and provide constructive feedback. We thank the reviewer for their time and helpful comments. We have done our best to incorporate reviewer suggestions into the manuscript or noted otherwise. Please find below our responses to reviewer comments.
Sincerely,
Ryan Schmid
Kelton Welch
Jon Lundgren
Reviewer 2
The Authors sampled the dung arthropod community in the Northern plains region (South Dakota, Minesota and North Dakota) three times (June, July and August/September) during two grazing seasons (2019 and 2020).
They characterized the arthropod communities calculating abundance, species richness, species diversity (Shannon H), species evenness (Shannon equitability), functional group abundance and functional group species richness.
Their main results are:
- They collected 51,290 arthropod specimens, belonging to 789 morphospecies, from 596 dung pats from 40 pastures
- Diptera, Coleoptera and Hymenoptera represent the majority of arthropod community abundance (94.5%) and species richness (77.8%).
- Arthopod abundance and species richness seems do not show a clear pattern during the three period of sampling.
- Arthopod diversity and evenness showed a steady decline from early to late sampling period.
- Coprophagous and predators represent the majority of the functional groups abundance and species richness
Overall, I think that the data of the authors are interesting, especially due to the large spatial scale of the sampling. However, I think the data could be presented better. Indeed, my major concern are about the data presentation and several speculative points in the discussion.
Some possible suggestions to improve the manuscript are:
- Line 56: you use the words “trophic groups”, “functional group”, “guild”…please be consistent. They are not synonyms.
Agreed. We have edited the manuscript throughout to specify “functional group”, as this is the term we intended to measure.
- Line 61: (Fig. 1)…. Where is Fig 1?
Figure 1 was accidently omitted during the formatting process for the journal. It has been inserted into the manuscript. (Line 69)
- Lines 90-91 and throughout the article and supplementary material: I think the authors have to change dwellers with non-nesters. Indeed, the category dweller include also endocoprid species which are not present in your area. I suggest to the authors to avoid the categories dwellers, tunnelers and rollers due to their ambiguity (especially regarding dweller category), and use the chategories telecoprid, paracoprid, endocoprid and non-nesters. I suggest to make this change throughout the manuscript. See Tonelli, M., 2021. Some considerations on the terminology applied to dung beetle functional groups. Ecological Entomology, 46: 772-776.
We agree with the reviewer that it is important to use the correct terminology when describing dung beetle functional groups, and there has been an evolution of this terminology in the literature that has led to some ambiguity during the last 50 years. However, it seems most of the literature has converged in agreement that dweller, tunneller, roller are acceptable terms when describing the function groups of dung beetles, and dweller, tunneller, and roller are interchangeable with endocoprid, paracoprid, and telecoprid, respectively. This is even highlighted in the Tonelli (2021) article referenced by the reviewer, where 913 of the 1143 articles (80%) included in the literature review used the terms dweller, roller, or tunneller to describe dung beetle functional groups. We agree with the literature on this point, as a functional group concerns how a resource or ecological component is processed by different species to provide a specific ecosystem service or function. In the case of dung beetles, classification by their resource allocation strategy according to the way they use and disrupt dung has led to the classification of dung beetles into the functional groups of dweller, tunneller, and roller. Therefore, we would prefer to keep the dung beetle functional groups described in this manuscript as dweller, tunneller, and roller, but we would be willing to substitute endocoprid, paracoprid, and telecoprid, if the reviewer would prefer these terms instead. However, we do not think endocoprid should be further divided to distinguish non-nesters from endocoprid because this distinction would not follow the majority of the literature.
That all being said, we would prefer this not become of major point of contention and prevent this manuscript from being published. Therefore, if the reviewer disagrees with our proposal, we would defer to the manuscript’s editor, and abide by whatever they think best.
- Results: I would like to see some measure of the amount of variation of your data… in the text (e.g. when you show the mean number of specimens per dung pat) and in the figure (e.g. figure 3)
Unfortunately, we are unable to show variation in the data as presented because it is not mean but total count data. However, we understand the value of the reviewer’s request, so we have edited Figure 3 to show the mean arthropod community metrics per pat, with variation (SEM) displayed on the graph. We have also included this data in the text. (Lines 131-153)
- Functional groups: you did not report the results regarding richness in the different functional groups.
This was an oversight on our part. We have added species richness data throughout section 3.2. Functional groups.
- Functional groups: if you want to have an ecological functional approach, I think it is of little use to continue dividing specific richness and abundance into different taxonomic orders. I think it is better to merge all the data of the different taxa using only the functional groups as categories.
We would like to maintain our current approach of presenting the functional groups divided up by taxonomic orders. We feel this is a useful organizationally for readers, as many studies of the dung arthropod community focus on specific taxonomic orders, e.g., dung beetles, dung feeding flies, pests, etc. This allows the manuscript to be a useful resource for readers interested in those specific groups. However, we understand the value in the reviewer’s comment, so we have added a paragraph at the end of this section that summarizes the functional groups collectively across the orders of the arthropod community. (Lines 231-234)
- Lines 154-155 (Figure 4): ….”citing literature and current hypotheses of the ecology of these organisms…” ????
We have removed the text in question because it was confusing and unnecessary for the reader to understand the figure caption. (Line 194)
- Figure 5: I think this is not the best way to show the variability of the data. Maybe a table with the variance of the data could be better.
We have overlaid line graphs of the means with variance represented by error bars. As seen in the revised figure, the error bars do not represent the variability in the data as seen in the bar graphs on the figure. Therefore, we would like to retain the bar graphs in the figure, unless there is a better suggestion. (Figure 5)
- Lines 203-204: generally Hydrophilidae are not considered “dung beetles”. (Krell, F.-T. & Moon, A.R. 2019. Quick guides: Dung beetles. Current Biology 29 (12): R554‒R555)
We agree with the reviewer that Hydrophilidae are not generally regarded as true “dung beetles”; however, they are an abundant dung feeding beetle that dwells within cattle dung in the region (Floate. 2011. “Arthropods in cattle dung on Canada’s grasslands”, Hanski and Cambefort. 1991. “Dung Beetle Ecology”). As such, they are an important part of the coprophagous functional group in the dung dwelling arthropod community and deserve to be recognized in this dataset. In order to not confuse readers by not lumping Hydrophilids in with the strict definition of “dung beetles”, but also recognize that Hydrophilids are an important component of the coprophagous beetle community, we have replaced the term “dung beetle” throughout the manuscript where appropriate and replaced it with “coprophagous beetles” or “dung feeding beetles”. We have also informed the reader that Hydrophilids are included with these terms (Line 103).
- Lines 235-255: this paragraph seems too speculative. For example, Hirschberger studied only Aphodius ater, but you extrapolate this data to all the Aphodiinae. Moreover, your data of 15.8 dweller/pat consider also predator Hydrophilidae (in the suppl material predator Hydrophilidae are categorized also as coprophagous and dwellers)?
We concur with the reviewer’s assessment. We have revised the paragraph substantially to focus on the known attributes of dung-feed beetle physiology and ecology that affect dung degradation rates. (Lines 255-350)
Lines 256-279: to make this part less speculative you could see if the pests are less in the dung pats where coprophagous fauna is present. Moreover, is true that insecticides can control pests, but it also has many negative effects on dung beetle biodiversity and their ability to degrade excrement (see for example Tonelli et al. 2020. Dung beetles: Functional identity, not functional diversity, accounts for ecological process disruption caused by the use of veterinary medical products. J. Insect Conserv., 24: 643–654. https://doi.org/10.1007/s10841-020-00240-4.
Verdù et al. 2018. Ivermectin residues disrupt dung beetle diversity, soil properties and ecosystem functioning: An interdisciplinary field study. Science of the Total Environment, 618: 219-228. https://doi.org/10.1016/j.scitotenv.2017.10.331)
We agree with the reviewer’s assessment of the paragraph. We have revised it to be less speculative and include the reviewer’s suggestions in combination with suggests from reviewer 1. (Lines 351-379)
- Supplementary Material: Colobopterus erraticus and Teuchestes fossor are paracoprid species (Rojewski, C., 1983. Observations on the nesting behaviour of Aphodius erraticus (L.). Polskie Pismo entomologiczne, 53: 271-279. ----- Zunino, M., Barbero, E., 1990. Food relocation and the reproductive biology of Aphodius fossor (L.) (Coleoptera Scarabaeidae Aphodiinae). Ethology Ecology & Evolution, 2: 334). Moreover, in the column EO the species name is missing.
We have changed the label of Colobopterus erraticus to tunneler; however, we were unable to confirm Teuchestes fossor was a tunneler in the citation provided by the reviewer because it is unavailable online. Furthermore, our sources label T. fossor as a dweller (Pokhrel et. al. 2021. “A review of dung beetle introductions in the Antipodes and North America: Status, opportuntiies, and challenges”. Cornell College of Agriculture and Life Sciences. https://cals.cornell.edu/new-york-state-integrated-pest-management/eco-resilience/beneficial-insects/visual-guide-dung-beetles/teuchestes-fossor). Thus, we have left T. fossor labelled as a dweller. We would be willing to change the label, if there is literature available that clarifies the situation.
Also, Column EO was T. fossor. Because we had already identified Scarabaeidae 008 (Column EI) to be T. fossor, we added the count data from Column EO to Column EI and deleted Column EO.
Round 2
Reviewer 2 Report
Comments and Suggestions for Authors
Congratulations for the great effort in considering my suggestions.
I think your article is much improved, more readable and less speculative.
However, I believe that a little further reflection on the functional groups
of dung beetles is necessary.
I don't agree that the choice of a terminology should be made on the basis of the quantity of articles that have, in the past, chosen that terminology.
It is true that the terms roller and tunneler are interchangeable with telecoprids and paracoprids, but it cannot be the same with dweller (it is absolutely not interchangeable with endocoprid) and equating trophic and reproductive behavior.
Hence, I strongly suggest that the authors change the terminology as proposed by Tonelli 2021 (non-nesters; telecoprid; paracoprid; endocoprid) in order to begin to avoid conceptual errors that can lead to ecological misinterpretations.
Moreover, I attached the paper about T. fossor reproductive biology.

Author Response
Reviewer comments: Round 2
We thank the reviewer for this round of comments. We have done our best to incorporate these ideas into the manuscript, and we believe it has clarified some issues in our manuscript for readers.
Thank you,
The authors (Ryan Schmid, Kelton Welch, and Jon Lundgren)
Reviewer 2 comments
Congratulations for the great effort in considering my suggestions.
I think your article is much improved, more readable and less speculative.
However, I believe that a little further reflection on the functional groups
of dung beetles is necessary.
I don't agree that the choice of a terminology should be made on the basis of the quantity of articles that have, in the past, chosen that terminology.
It is true that the terms roller and tunneler are interchangeable with telecoprids and paracoprids, but it cannot be the same with dweller (it is absolutely not interchangeable with endocoprid) and equating trophic and reproductive behavior.
Hence, I strongly suggest that the authors change the terminology as proposed by Tonelli 2021 (non-nesters; telecoprid; paracoprid; endocoprid) in order to begin to avoid conceptual errors that can lead to ecological misinterpretations.
We understand the reviewer’s concern about the correct use of terminology for dung beetle functional groups, we want to get this correct too. Our concern is that we have not seen the distinction made between endocoprids and non-nesters in peer-reviewed literature by other authors over the last several decades besides in Tonelli papers. Meanwhile dwellers, rollers, and tunnelers are still commonly used in the peer-reviewed literature and by the public to describe dung beetle functional groups, and we would like to remain consistent so as not to introduce confusion with peers and the public. However, to accommodate the reviewer’s concerns we have specifically defined our use of the term dweller in the manuscript, referencing the distinction in terminology proposed by Tonelli (2021) and recognizing this difference when discussing the results of the dwellers collected in this survey, stating the breakdown of non-nesters and endocoprids in our dweller categorization. (Lines 108-117, 197, 313)
Moreover, I attached the paper about T. fossor reproductive biology.
We thank the reviewer for providing the paper that we did not have access to. After reviewing this paper, it seems there is some conflicting data about T. fossor in the literature. Our sources say that T. fossor is a dweller [1, 2, 3], with Gittings and Giller [1] observing the oviposition and larval development of T. fossor in dung pats in the field. Thus, we are unsure how to classify T. fossor. Our proposed solution is below, but we will differ to the editor’s discretion to rectify this problem.
In order to not mislead readers on this subject we have referenced in the results section that there are conflicting reports on the classification of T. fossor in the literature. We also explained why we kept T. fossor labeled as a dweller in our results, stating that Zunino and Barbero (1990) acknowledge that the third larval stage develops in the dung pat over a longer period than the first and second larval stages in the soil. Also, we have stated the abundance of T. fossor in the results so that readers may determine how it would affect the discussion if it were to be classified as a tunneler. (Lines 198-205)
Our sources:
- Gittings and Giller. 1997. Cornell College of Agriculture and Life Sciences. Ecography 20: 55-66.
- Pokhrel et. al. 2021. A review of dung beetle introductions in the Antipodes and North America: Status, opportuntiies, and challenges. Environmental Entomology 50: 762-780.
- https://cals.cornell.edu/new-york-state-integrated-pest-management/eco-resilience/beneficial-insects/visual-guide-dung-beetles/teuchestes-fossor